# REAL2CODE: RECONSTRUCT ARTICULATED OBJECTS VIA CODE GENERATION

**Zhao Mandi**[1], **Yijia Weng**[1], **Dominik Bauer**[2], **Shuran Song**[1]
[1] Stanford University [2] Columbia University

## ABSTRACT

We present Real2Code, a novel approach to reconstructing articulated objects via code generation. Given visual observations of an object, we first reconstruct its part geometry using image segmentation and shape completion. We represent these object parts with oriented bounding boxes, from which a fine-tuned large language model (LLM) predicts joint articulation as code. By leveraging pre-trained vision and language models, our approach scales elegantly with the number of articulated parts, and generalizes from synthetic training data to real world objects in unstructured environments. Experimental results demonstrate that Real2Code significantly outperforms the previous state-of-the-art in terms of reconstruction accuracy, and is the first approach to extrapolate beyond objects' structural complexity in the training set, as we show for objects with up to 10 articulated parts. When incorporated with a stereo reconstruction model, Real2Code moreover generalizes to real-world objects, given only a handful of multi-view RGB images and without the need for depth or camera information. [1]

## 1 INTRODUCTION

The ability to reconstruct real-world objects in simulation (real-to-sim) promises various downstream applications: automating asset creation for building VR/AR experiences, enabling embodied agents to verify their interaction in simulation before execution in the real world (Lim et al., 2022; Wang et al., 2023a; Torne et al., 2024), or building large-scale simulation environments that support data-driven policy learning (Katara et al., 2023). We are particularly interested in articulated objects, for both their ubiquity in household and industrial settings and the unique challenges they pose in contrast to single-body rigid objects. To reconstruct articulated objects, prior learning-based methods typically train supervised (Jiang et al., 2022b) or test-time-optimized (Liu et al., 2023a) models on synthetic objects with simple articulation structures (i.e., one or two moving parts per object). This results in limited generalization ability to objects with more complex visual appearances and kinematics. In addition, prior work only provides object part reconstructions of limited quality: the extracted meshes are often incomplete and the predicted articulation parameters require manual cleanup before being usable for simulation.

We propose **Real2Code**, a novel approach to address the above limitations. We represent object articulation with code programs, and use language modeling to predict these code programs from visual observations. This formulation scales elegantly with objects' structural complexity: to process an articulated object with multiple joints, prior methods would require either changing the output dimension of their articulation prediction model, or run multiple inferences on pairs of before- and after- interaction observations to predict one joint at a time. In contrast, the next-token prediction formulation in language modeling allows generating arbitrary-length outputs, i.e., the model architecture needs no adjustment to handle varying number of object joints. Whereas prior work on shape programs (Tian et al., 2019) needs to define task-specific code syntax, we represent objects with simulation code in Python, which takes advantage of recent progress in large language models (LLMs) that are pre-trained with code generation capabilities.

Although capable at code generation, LLMs pre-trained on text are not as equipped at predicting accurate numerical values from spatial geometry information, which is required in our task in order

---

[1]Project Website: `https://real2code.github.io/`

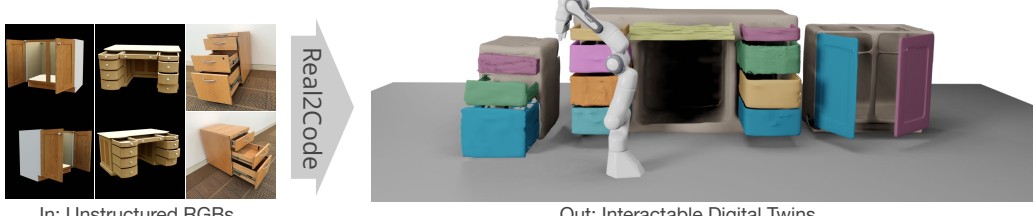

Figure 1: We propose a novel method for reconstructing articulated objects via code generation leveraging pre-trained large language models (LLMs). Real2Code takes visual observation input and performs both part-level geometry reconstruction and joint prediction. When evaluated on an extensive set of real and synthetic objects with varying levels of kinematic complexity (up to 10 parts), Real2Code demonstrates high reconstruction accuracy and generalizes to real world objects from a handful of pose-free RGB images.

to obtain articulated joint parameters. To address this, we propose to use oriented bounding boxes (OBBs) as an abstraction layer that summarizes the raw sensory observation to the LLM in a concise yet precise manner. Given partial observations of an object, we first perform part-level segmentation and reconstruction via a combination of 2D segmentation and a 3D shape completion model; next, OBBs are extracted from the reconstructed object parts, and serve as input to the LLM. The LLM then predicts joints as a classification problem by selecting the closest OBB rotation axis and box edges.

In unstructured real world environments, another challenge is the lack of accurate depth and camera information. To address this, we incorporate a pre-trained dense stereo reconstruction model, namely DUSt3R (Wang et al., 2023b), into our pipeline: we show the dense 2D-to-3D point-map prediction from DUSt3R can be combined with our fine-tuned SAM model to achieve view-consistent 3D segmentation. As a result, Real2Code can then reconstruct real world objects from only a handful of pose-free RGB images.

For a more systematic evaluation, we validate the performance of Real2Code on the well-established PartNet-Mobility dataset (Mo et al., 2019), using an extensive test set of unseen objects that contain various numbers of articulated parts. Compared to the prior state of the art, Real2Code significantly improves both 3D reconstruction and joint prediction accuracy. Real2Code is the only approach to reliably reconstruct objects with more than three articulated parts, whereas prior methods fail completely on such objects. Fig. 1 highlights our results on both synthetic multi-part objects (first column), where we show Real2Code reconstructs both synthetic objects with up to 10 parts (first column) and real-world objects (second column) using in-the-wild RGB images.

In summary, our contributions are threefold:

1. We present Real2Code, a novel approach to reconstructing articulated objects from a handful of unstructured RGB images. We formulate joint prediction as a code generation problem and adapt pre-trained large language models to specialize in this task.

2. We address part reconstruction via kinematic-aware view-consistent image segmentation and a learned 3D shape completion model, which leads to high-quality mesh extraction that generalizes to multi-part real-world objects.

3. Empirical results demonstrate that Real2Code significantly outperforms the prior state of the art at both articulation estimation and part reconstruction. To the best of our knowledge, Real2Code is the first method to accurately predict objects with more than three parts, and generalizes beyond the training dataset with at most 7 parts to objects with up to 10 parts.

## 2 RELATED WORK

**LLMs for Visual Tasks.** Pre-trained LLMs have been used for visual reasoning and grounding tasks (Zeng et al., 2022; Hsu et al., 2023b). LLMs' code-generation capability has also been exploited for generating programs that solve visual tasks (Gupta & Kembhavi, 2022; Surís et al., 2023; Subramanian et al., 2023). These works use zero-shot pre-trained LLMs such as GPTs (Brown et al., 2020; OpenAI, 2023) and require prompt engineering, such as providing in-context examples,

to guide the model to generate desirable outputs; in contrast, we directly fine-tune the weights of a code-generation model to specialize in our articulation prediction task without prompt tuning.

**Shape Programs.** Code-like programs have been studied in computer vision as a compact representation for 2D and 3D shapes. A main challenge for learning code programs is the lack of supervision, and prior work has explored using learned differentiable code executor (Tian et al., 2019), pseudo-labeling (Jones et al., 2022), differentiable rendering (Liang, 2022), imitation learning on code sequences (Willis et al., 2021), or reinforcement learning (Tulsiani et al., 2016). More recent work has explored constructing large-scale datasets of shapes (Ganin et al., 2021) or scene layouts (Avetisyan et al., 2024) and train supervised LLM-like models to generate code outputs. In contrast to ours, these prior works focus on either individual object shapes or scene-level room layouts, but do not estimate joint articulations. In addition, instead of the task-specific code programs, such as customarily-designed language syntax (Tian et al., 2019; Jones et al., 2022; Avetisyan et al., 2024) or Computer-Aided Design (CAD) code (Willis et al., 2021; Ganin et al., 2021), we represent object articulation with Python code that 1) closely matches the pre-training distribution of code-generation LLMs, which allows fine-tuning with limited data, and 2) can be directly executed by a physics simulator (Todorov et al., 2012), which makes the reconstruction more usable for simulation and requires less manual cleanup.

**Articulation Model Estimation.** Prior work has investigated estimating pose and joint properties of articulated objects *without* full reconstruction. A common setup is to assume physical interactions on an object to infer its articulation information: classical sampling-based algorithms (Huang et al., 2014; Katz et al., 2013) are proposed to estimate joint parameters based on sensory inputs from an object's different configuration states; other learning-based methods train end-to-end models to predict part-level segmentation, kinematic structure, object part poses, or articulated joint parameters (Hu et al., 2017; Yi et al., 2018; Wang et al., 2019; Michel et al., 2015; Li et al., 2020; Zeng et al., 2021; Huang et al., 2021; Tseng et al., 2022; Abdul-Rashid et al., 2022; Jiang et al., 2022a; Liu et al., 2023b). Buchanan et al. (2023); Heppert et al. (2022); Sun et al. (2023) propose specialized neural network architectures to improve the estimation performance. Other works focus on learning to propose the most informative physical interactions on an object to help robot manipulation (Mo et al., 2021), or to better isolate and segment articulated parts and joints (Gadre et al., 2021). These articulation estimation tasks provide useful metrics for 3D shape reasoning (Wang et al., 2019), and Liu et al. (2022); An et al. (2023); Geng et al. (2023b;a) show that the predicted object pose and joint information are useful for robotic tasks. However, prior work typically handles objects with simple structure (i.e., one or two moving parts) and does not address full object reconstruction. In contrast, our method handles objects with more than ten moving parts, and performs shape reconstruction via extraction of part meshes.

**Articulated Object Reconstruction.** Most closely related to ours are methods that reconstruct both the geometry and joints of articulated objects. A popular approach is to train end-to-end models on synthetic data to jointly segment articulated parts and predict joint parameters, assuming either observations from interactions (Jiang et al., 2022b; Hsu et al., 2023a; Nie et al., 2023; Mu et al., 2021) or single-stage (Heppert et al., 2023; Irshad et al., 2022; Kawana et al., 2022; Wei et al., 2022) observations. Another approach uses per-object optimization (Liu et al., 2023a;b) without training. Based on observations of the object in two or more different joint states, it optimizes for joint parameters to match observed motion correspondences and optionally performs 3D reconstruction using learned neural fields. Most existing methods assume a single joint and do not scale well with an increasing number of joints: for example, to handle an object with $N$ joints, methods like Ditto (Jiang et al., 2022b) would need to move the $N$ joints one by one, record the observations before and after each interaction, and run $N$ inferences on each observation pair. PARIS (Liu et al., 2023a) would need to optimize $N$ neural fields and joint parameters, which may lead to a much more complex optimization landscape. The approach presented by Liu et al. (2023b) handles multiple joints but requires a complete sequence of point-cloud observations and is not able to reconstruct 3D shapes.

## 3 METHOD

We address the problem of reconstructing multi-part articulated objects from visual observations. An articulated object is composed of a set of rigid-body parts that are connected via joints. We assume that joint types are either prismatic or revolute: a prismatic joint is parameterized by a joint axis $\mathbf{u}^p \in \mathbb{R}^3$ and a translation offset $d$; a revolute joint is parameterized by a position $\mathbf{p}^r \in \mathbb{R}$, a rotation axis $\mathbf{u}^r \in \mathbb{R}^3$, and a rotation angle $\theta$. For an object with $N$ moving parts, we assume each to be

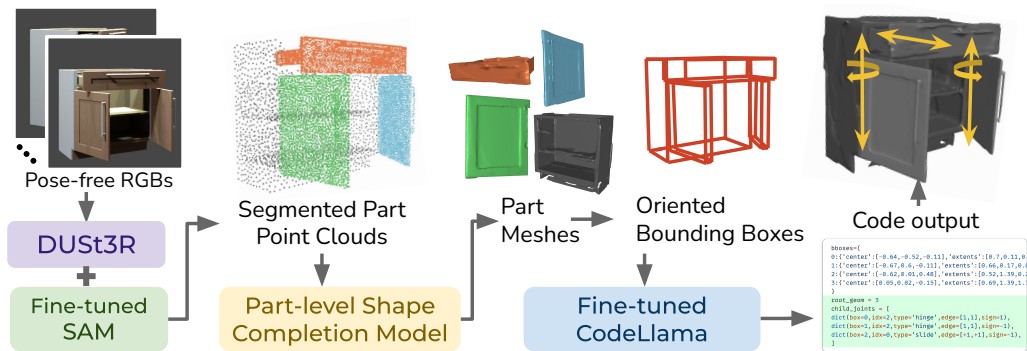

Figure 2: Overview of Real2Code pipeline. Given unstructured multi-view RGB images, we leverage the pre-trained DUSt3R model (Wang et al., 2023b) to obtain dense 2D-to-3D pointmaps, and use a fine-tuned 2D segmentation model(Kirillov et al., 2023) to perform part-level segmentation and project to segmented 3D point clouds. We train a shape-completion model to take partial point cloud input and predict a dense occupancy field, which is used for part-level mesh extraction. We fine-tune a large language model (LLM) (Rozière et al., 2023) that takes mesh information in the form of oriented bounding boxes, and outputs full code descriptions of the object that can directly be executed in simulation.

connected with its parent via exactly one 1-DoF joint. Therefore, the transformation between each part's frame and its parent's frame is uniquely determined by the joint parameters. Furthermore, for hinge joints, we focus on objects what have joint position lie closely with one of its oriented bounding box (OBB) edges — we remark that this is true for many common household objects with cuboid shapes, such as doors, boxes, laptops, etc.

To obtain visual inputs, we assume an object is manipulated such that each articulated joint is at a non-zero state, i.e., $d > 0$ or $\theta > 0$, when we capture multi-view RGB (and optionally depth) images. Our system outputs a set of 3D meshes – each a reconstruction of the object's parts – and a list of joint types and parameters represented as code. The outputs can then be used to create the object's digital twin in simulation for downstream applications.

Fig. 2 provides an overview of our method. Real2Code consists of two main steps: reconstruction of object parts' geometry (described in Sec. 3.1) and joint estimation via LLM code generation (described in Sec. 3.2). Between the two steps, the oriented bounding boxes (OBBs) of the object parts serve as an abstraction layer, enabling the LLM to reason about 3D spatial information and predict accurate joint parameters.

## 3.1 PART RECONSTRUCTION

To reconstruct an object's part-level shapes, we propose a 2D-to-3D approach that is category-agnostic and handles objects with an arbitrary number of parts. First, we fine-tune a SAM model that generates 2D segmentations from RGB images, and project them to partially-observed 3D point clouds. Next, we train a shape completion model that takes 3D point cloud input and extracts watertight meshes.

### 3.1.1 KINEMATICS-AWARE PART SEGMENTATION

We leverage the pre-trained 2D segmentation model SAM (Kirillov et al., 2023) to segment object parts based on their kinematic structure. This design is motivated by the need to 1) generalize to real world data, and 2) scalability to the number of object parts. In contrast to prior works that train 3D segmentation models using limited amount of synthetic data (Jiang et al., 2022b; Mo et al., 2019; Xiang et al., 2020), SAM (Kirillov et al., 2023) was pre-trained on a much larger dataset. Therefore, SAM generalizes better to in-the-wild real world images, with a strong prior to identify moving object parts without the need for multi-step interactions.

However, because SAM (Kirillov et al., 2023) is originally designed for iterative user prompting, its zero-shot predictions do not always match the articulation structure, e.g., segmenting unnecessary details on an object part. To address this, we fine-tune the pre-trained model using PartNet-Mobility (Mo et al., 2019; Xiang et al., 2020) data: the model's heavy-weight image encoder is kept frozen, we update the lightweight prompt-decoder layer to take an image and one sampled 2D point prompt as input, and predict the correct corresponding mask. See appendix A.3 for more details.

### 3.1.2 TEST-TIME PROMPTING FOR VIEW-CONSISTENT SEGMENTATION.

The point-based segmentation described above scales easily with the number of the object parts. However, this formulation also inherently lacks view consistency, as SAM is unaware of the correspondences across different camera views. To address this, we introduce a test-time prompting procedure to project predicted 2D masks into a view-consistent 3D segmentation. We discuss two different input settings based on the availability of depth and camera data: **1) Multi-view RGB-D and Camera Input**: we first coarsely sample 2D points on each RGB image and run the SAM model to obtain the background masks. This

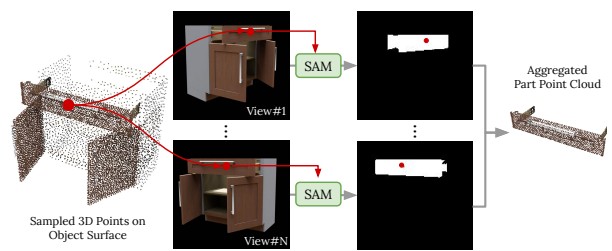

Figure 3: **View-consistent segmentation.** Illustration of our method for test-time prompting the fine-tuned SAM model. We first sample 3D points from the foreground object point clouds, project each point onto 2D RGB images captured from different camera views, which are used to prompt the model to generate view-consistent segmentations.

allows us to segment the foreground object in the different views and sample 3D points uniformly on the object point cloud. Next, we project each such 3D point back onto each image, and obtain view-consistent 2D points for SAM prompting. Further, we rank the model's predicted masks based on the confidence and stability scores proposed by Kirillov et al. (2023), and filter them using non-maximum suppression (NMS) to produce the final 3D segmentation. **2) Multi-view Unstructured RGB Input.** To handle real world settings which often lack high-quality depth and camera information, we adopt a multi-view stereo reconstruction model to achieve part segmentation. We use the recently proposed DUSt3R (Wang et al., 2023b) model, which is pre-trained to predict dense 3D point-maps from RGB input images. We then sample 2D points from one RGB image and find each point's corresponding point in every other RGB images via nearest-neighbor. More details are described in appendix A.4. This overall procedure of projecting between 3D to 2D prompting is similar to SA3D (Cen et al., 2023), which samples on a NeRF(Mildenhall et al., 2020) field and uses inverse rendering to effectively prompt SAM in 3D.

### 3.1.3 PART-LEVEL SHAPE COMPLETION.

Due to frequent self-occlusion, e.g. the inside of a drawer is often not visible, RGB-D input does not provide full observation of each object part, and subsequently a segmented point cloud does not recover complete part shape. This motivates learning a shape completion model to obtain watertight meshes. Because part-segmentation is already handled in the previous step, we here tackle shape completion on the object part level. We build on top of Convolutional Occupancy Nets (Peng et al., 2020): the model architecture consists of a PointNet++(Qi et al., 2017) point-cloud encoder, followed by a 3D Unet (Özgün Çiçek et al., 2016) encoder and a linear MLP decoder that predicts logits for occupancy. We use the ground-truth part meshes from PartNet-Mobility (Mo et al., 2019) to generate a dataset of partial point cloud inputs and occupancy labels. We normalize the occupancy grid using *partial* OBBs extracted from the input point cloud to avoid under-fitting the smaller-sized meshes. Marching Cubes (Lorensen & Cline, 1987) is used to extract the completed part meshes from predicted occupancy. See appendix A.3 for more details.

### 3.2 ARTICULATION PREDICTION VIA CODE GENERATION

Given a set of segmented object parts, we next predict their articulation structure using LLM-based code generation. This approach yields several advantages: first, code offers a compact representation for joints, and when combined with LLM's ability to predict arbitrary-length outputs, it scales elegantly with the complexity of object kinematic structure; second, pre-trained LLMs are equipped with strong priors for both common-sense objects and for generating syntactically correct code, making them easily adaptable to our task; lastly, the LLM-generated code can be directly executed in simulation, removing the need for manual cleanup of predicted joint parameters as seen in prior work (Jiang et al., 2022b). The following sub-sections first introduce our formulation of predicting joint parameters from oriented bounding boxes, then discuss our LLM fine-tuning procedure.

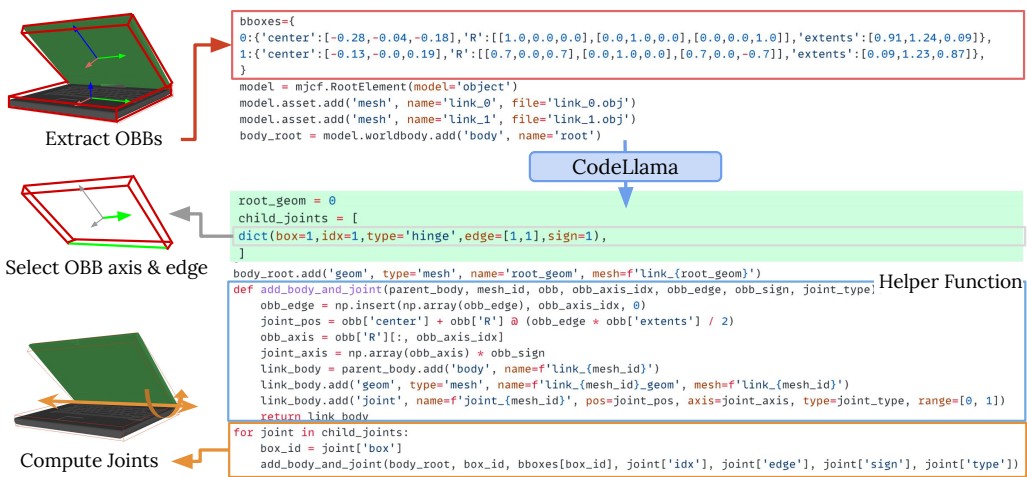

Figure 4: **Articulation Prediction as Code.** We fine-tune a Codellama (Rozière et al., 2023) model that takes in oriented bounding boxes (OBBs) for segmented parts as input, and generates joint predictions via selecting OBB rotation axes and edges (model generation is highlighted in green). A helper function is used to compute the absolute joint axis and position that assembles the object parts in simulation

### 3.2.1 ORIENTED BOUNDING BOX AS INPUT ABSTRACTION.

Articulation prediction requires numerical precision at joint parameters (i.e., position and rotation) and reasoning from raw sensory input, but an LLM pre-trained on text is not adept at these challenges. We address this by representing the sensory input (object point clouds) as a set of oriented bounding boxes (OBBs), each representing a segmented and completed object part. Compared to alternative object representations such as 3D point clouds or 2D images, OBBs strike a balance between **compactness** (i.e., do not require an extra feature encoder) and **preciseness** (i.e., provide numerical 3D pose information). Further, OBBs provide a reference for joint information. Recall that the pose of an object part is determined by its 1-DoF joint at a non-zero state — we can hence recover joint parameters from the observed displacement of object parts. Given an OBB of a part connected to its parent, the joint axis will be parallel to one of the three axes of the OBB's rotation matrix regardless of its joint type. We observe many common articulated objects consist of cuboid-like parts (e.g. doors or laptops), hence the position of their corresponding revolute joints will lie closely to, if not overlap with, one of the OBB edges. Combining these observations, we re-formulate the joint axis prediction problem by selecting an OBB rotation axis as the joint axis and, for revolute joints, choosing an OBB edge parallel to the axis as the joint position. See Fig. 4 for an illustration.

### 3.2.2 FINE-TUNING A CODE GENERATION LLM.

We now have an input formulation that effectively converts a regression task (i.e., predicting continuous values) to an easier classification task (i.e., selecting axes and edges) for LLMs. We use the 7B-CodeLlama (Rozière et al., 2023) model for its open-source-availability and built-in priors for code generation. We construct a fine-tuning dataset using PartNet (Mo et al., 2019) objects (the same assets used to generate our segmentation and shape completion data), and convert the native URDF files into MJCF code (Tunyasuvunakool et al., 2020), which 1) is in the more compact Python syntax, 2) can be executed in MuJoCo (Todorov et al., 2012) physics simulation, and 3) has each object's joints assigned with respect to the corresponding part's OBB information. The LLM takes a list of part-OBB information (i.e., center, rotation, and half-lengths) as input, and outputs joint predictions as a list, where each line contains indices into the axes and edges of an OBB. More details can be found inappendix A.3.

## 4 EXPERIMENTS

We evaluate Real2Code and compare to baseline methods to validate the effectiveness of our approach. Sec. 4.2 describes experiments on our kinematics-aware 2D image segmentation and 3D shape completion models. Sec. 4.3 evaluates our fine-tuned code-generation model on articulation prediction.

| Category | Laptop | | Box | | Fridge | | Furniture | | Furniture | |
| Number of Parts | 2 | | 2 | | 2-3 | | 2-4 | | 5-15 | |
| Metric | whole | part | whole | part | whole | part | whole | part | whole | part |
| --- | --- | --- | --- | --- | --- | --- | --- | --- | --- | --- |
| Real2Code+gtSeg | 0.57 | 2.33 | 1.54 | 7.65 | 0.51 | 2.04 | 1.46 | 13.3 | 5.84 | 16.8 |
| Ditto | 2.54 | **2.04** | 1.73 | 82.82 | 2.80 | 462.25 | **2.25** | 1105.86 | **2.21** | 4608.08 |
| PARIS | 84.29 | 206.31 | 15.35 | 158.73 | 20.63 | 1297.27 | 6.02 | 544.64 | 11.44 | 816.86 |
| Real2Code-SegOnly | 1.74 | 7.19 | 11.46 | 10.52 | 0.90 | 23.44 | 17.43 | 206.49 | N/A | N/A |
| Real2Code (Ours) | **0.44** | 3.02 | **1.31** | **5.94** | **0.60** | **1.28** | 3.47 | **65.79** | 19.70 | **118.58** |

Table 1: We evaluate surface reconstruction quality by measuring Chamfer-Distance (CD) between predicted and ground-truth meshes. Results are reported separately for each object category, where we take average CD of objects' entire surface reconstruction ('whole' column) and of all part wholes ('part' column). Objects from Storage-Furniture and Table are reported under Furniture and divided based on the number of parts.

Sec. 4.5 contains ablation studies that provide additional insights into our method. Sec. 4.6 shows qualitative results of our pipeline on real world objects.

## 4.1 EXPERIMENT SETUP

**Datasets.** We use assets from five categories in PartNet-Mobility (Mo et al., 2019) dataset: Laptop, Box, Refrigerator, Storage-Furniture and Table. The same split of 467 train and 35 test objects are used to construct our image segmentation, shape completion, and code datasets. We use Blender (Community, 2018; Denninger et al., 2023) to render RGB-D and segmentation masks. The RGB-D images and masks are then used to generate part-level point clouds as partial observations. For code data, we extract OBBs from part meshes and process each object's raw URDF file into Python MJCF (Tunyasuvunakool et al., 2020), where the joint rotation and position are relative to the OBB of the child part that this joint connects to the parent part. Refer to appendix A.2 for more details.

**Baselines.** We compare Real2Code to the following baseline methods:

• **PARIS** (Liu et al., 2023a) is the prior state-of-the-art for articulated object reconstruction. It takes multi-view RGB observations of a two-part articulated object at two different joint states, then optimizes NeRF-based reconstructions and joint parameters based on motion cues from the two observed states. We render our test objects at two random joint states, report the average performance across 5 random initialization seeds. We modify their method to optimize for more than two parts at once. However, we observe that their design of optimizing one neural field for each part runs out of memory when the number of joints exceeds 4.

• **Ditto** (Jiang et al., 2022b) is an end-to-end learned model that takes in a pair of before- and after-interaction point cloud inputs and predicts implicit part shapes and joint parameters. Notably, Ditto assumes only one object part is moved at a time, which requires step-by-step interactions and observations, making evaluation less efficient. For an object with $N$ joints, we move one part at a time, render the corresponding $N$ pairs of point cloud observations, and run their pre-trained model $N$ times to obtain the final results.

• **GPT-4** (OpenAI, 2023) is representative of recent state-of-the-art LLMs with strong reasoning and code-generation capability. We use it as a reference for zero-shot LLM performance on our task without fine-tuning. We prompt it with the same code header used in our LLM fine-tuning dataset, plus additional instructions on how to format the output, which our fine-tuned LLM does not need.

## 4.2 PART SEGMENTATION AND RECONSTRUCTION EXPERIMENTS

### 4.2.1 3D PART-LEVEL SHAPE COMPLETION.

Following the prompting procedure described in Sec. 3.1, we first run our fine-tuned SAM on images from the test set of unseen objects and obtain segmented part point clouds. We observe that, because we rank and filter the mask predictions (i.e., prioritize high predicted confidence score and stability score), the low-quality masks have less impact on the final segmented point-cloud after the projection step. Next, we use the segmented part point clouds as input to evaluate our learned shape completion model: following Mu et al. (2021); Jiang et al. (2022b); Liu et al. (2023a), we uniformly sample $10,000$ points on the extracted mesh surface, and report the average Chamfer Distance between the extracted and ground-truth part meshes in Tab. 1. Because the predictions are semantics-agnostic

| | 2 Parts (15) | | | 3 Parts (9) | | | 4-5 Parts (6) | | | 6-15 Parts (7) | | |
|---|---|---|---|---|---|---|---|---|---|---|---|---|
| | rot↓ | pos↓ | type↑ | rot↓ | pos↓ | type↑ | rot↓ | pos↓ | type↑ | rot↓ | pos↓ | type↑ |
| Real2Code+gtBB | 0.0 | 0.07 | 0.93 | 0.0 | 0.04 | 1.00 | 0.0 | 0.04 | 1.00 | 11.6 | 0.03 | 0.94 |
| Ditto | 40.04 | 4.04 | 0.57 | 35.57 | 2.47 | 0.70 | 49.77 | 3.20 | 0.43 | 63.06 | 4.16 | 0.30 |
| PARIS | 48.44 | 2.67 | 0.51 | 32.35 | 3.63 | 0.84 | 55.97 | 2.14 | 0.43 | N/A | N/A | N/A |
| GPT4 | 57.3 | 0.26 | 0.73 | 10.0 | 0.08 | 0.61 | 45.0 | 0.21 | 0.51 | **30.0** | **0.05** | 0.71 |
| Real2Code (Ours) | **7.5** | **0.08** | **0.80** | **0.0** | **0.04** | **0.89** | **0.63** | **0.07** | **0.97** | 30.2 | **0.05** | **0.89** |

Table 2: Joint prediction results from Real2Code and baseline methods, grouped by the number of moving parts in each object. We remark that Real2Code consistently outperforms baseline methods across objects with different kinematic structures; on objects with 4 or more moving parts, Real2Code predicts joints accurately whereas baseline methods fail.

(i.e., the model does not know if a segmented part is a drawer or a door), we generate permutations of the set of predicted meshes and take the permutation that results in lowest error; the same logic is used for joint prediction results.

Overall, Real2Code achieves the best reconstruction quality and elegantly scales to a larger number of parts (column 'Real2Code (Ours)'). We remark on the performance difference between Real2Code and baselines: the joint optimization of all parts in PARIS (Liu et al., 2023a) suffers from a complex loss landscape and produces unsatisfactory reconstructions, especially when the number of parts increases. Ditto (Jiang et al., 2022b) performs well on training categories (i.e., Laptop) but does not generalize well to unseen categories. In contrast, ours obtain better results because we factorize the problem into segmentation and shape completion, aggregate 2D segmentation from fine-tuned SAM and perform part-level shape completion.

To validate the need for our shape completion model, we observe that 1) Due to the partial observation and noise in the segmentation masks, simply extracting meshes from the part-level point clouds also results in subpar reconstruction results (column 'Real2Code-SegOnly', where 'N/A' indicates the mesh extractions are too noisy to match with GT mesh). 2) If we use ground-truth segmentation, the mesh extraction from the aggregated point clouds are better than using SAM segmentation, but are still incomplete (column 'Real2Code-gtSeg').

### 4.2.2 KINEMATICS-AWARE 2D IMAGE SEGMENTATION.

To demonstrate the effectiveness of SAM fine-tuning, we evaluate the fine-tuned model on unseen object images by uniformly sampling a grid of $32 \times 32$ query points and compare the predicted segmentation with ground-truth masks. We use NMS filtering on the predicted masks, then sort with the model's predicted confidence score to take the top-K masks that fill the image to more than 95% total pixels. We observe a significant improvement over zero-shot SAM: object parts are segmented much more closely following their kinematics structure, obtaining a 92% mean IoU score on the final used masks and 84% match rate to ground-truth masks.

### 4.3 ARTICULATION PREDICTION EXPERIMENTS

After completing part-level reconstruction on test objects, we extract OBBs for each object part and compose a text-prompt for our fine-tuned CodeLlama model. We parse the model's code generation and append it with code header lines (e.g. import packages) such that the post-processed code can be directly executed to produce object simulation. We then evaluate the accuracy of articulation prediction by measuring the error of joint type, joint axis, and (for revolute joints only) joint position predictions. As shown in Tab. 2, Real2Code outperform all baseline methods by a large margin. The effectiveness of our OBB abstraction is further accentuated by column 'Real2Code+gtBB', where we feed oracle OBB to the code generation module and achieve highly accurate predictions even on unseen objects with a large number of parts.

### 4.4 QUALITATIVE RESULTS

For qualitative results, we show objects with a range of varying kinematic complexities, from a two-part laptop to a ten-part multi-drawer table. We visualize the final reconstructed objects from ours and baselines methods in Fig. 6. Whereas all methods handle the simpler laptop articulation, baseline methods struggle as the number of object part increases while Real2Code performs much more accurate reconstruction. See our submission website for more visualizations: `https://sites.google.com/view/real2code-submission`

| Inp. | Out | 2 Parts (15) | | | 3 Parts (9) | | | 4-5 Parts (6) | | | 6-15 Parts (6) | | |
|------|-----|------|------|-------|------|------|-------|------|------|-------|------|------|-------|
| | | rot↓ | pos↓ | type↑ | rot↓ | pos↓ | type↑ | rot↓ | pos↓ | type↑ | rot↓ | pos↓ | type↑ |
| OBB | Abs. | 7.5 | **0.06** | 0.92 | **0.0** | **0.03** | 1.0 | **0.0** | 0.6 | 0.83 | **0.0** | 0.7 | 0.73 |
| OBB | Rot. | 0.0 | 0.18 | 0.73 | 0.3 | 0.23 | 1.00 | 0.9 | 0.19 | 0.83 | 5.9 | 0.06 | 0.59 |
| +RGB | Rel. | **0.0** | **0.06** | 0.80 | 5.0 | **0.03** | 1.0 | **0.0** | **0.03** | 0.89 | **0.0** | **0.02** | 0.67 |
| OBB | Rel. | **0.0** | 0.07 | **0.93** | **0.0** | 0.04 | **1.0** | **0.0** | 0.04 | **1.00** | 11.6 | 0.03 | **0.94** |

Table 3: Joint prediction results from ablation experiments on Real2Code. Using a regression formulation, the LLM is still able to output reasonable values for two or three part objects, but generates much less accurate joint positions when the number of articulated parts increase. Additional RGB image input yields no clear improvements, which suggests the OBB input alone can provide sufficient information.

## 4.5 ABLATION STUDIES

To further validate our formulation of using OBB as reference for articulation prediction, experiment with alternative input and output representation for ablation:

- **Regression on Joint Parameters.** We fine-tune two more CodeLlama models to take the same input but outputs continuous numerical values for joint parameters: the first model directly predicts 3 values for each joint axis and 3 for every joint position (Sec. 4.5 row 'OBB Abs.'); the second model predicts joint axis the same way as Real2Code, but predicts joint position as a relative position to the OBB's center (Sec. 4.5 column 'OBB Rel.').

- **Provide LLM with Visual Inputs.** We fine-tune a model with both RGB and OBB inputs. We adopt the OpenFlamingo (Alayrac et al., 2022; Awadalla et al., 2023) approach for interleaving image embeddings with the CodeLlama model weights, and uses the same pre-trained ViT (Dosovitskiy et al., 2020) weights for image encoder.

Results from the ablation experiments are reported in Tab. 3. We make the following remarks: first, regression formulation predicts less accurate joint positions. Both predicting absolute joint positions (column 'OBB Abs.') and relative position from OBB center (column 'OBB Rot.') yield a higher error. In contrary, the rotation error is still on a reasonable scale: we found this is due to the model learns to copy the correct axis column from the OBB rotation matrices contained in the input prompt. Second, RGB input does not yield significant improvement. Comparing row '+RGB Rel.' and 'OBB Rel.', we see the OBB input provides sufficient information for articulation prediction task.

## 4.6 EXPERIMENTS ON REAL WORLD OBJECTS

To validate the generalization ability of Real2Code, we gather a set of in-the-wild articulated objects and collect multi-view RGB data as inputs. We run Real2Code with DUSt3R (Wang et al., 2023b) to achieve reconstruction from multi-view pose-free RGB images. Due to the lack of ground-truths, we show qualitative results in Fig. 7 that Real2Code generalizes well to real objects and produces good quality reconstructions from RGB-only inputs. However, although the learned DUSt3R (Wang et al., 2023b) model performs well on overall shape and exterior surface areas of the objects, it predicts less accurate point maps at areas inside the drawers, (likely due to the lack of similar data in their training dataset). As a result, the segmented part point clouds display noises (second row in Fig. 7), and leads to lower quality mesh extractions. See appendix A.4 for more details on our evaluation setup.

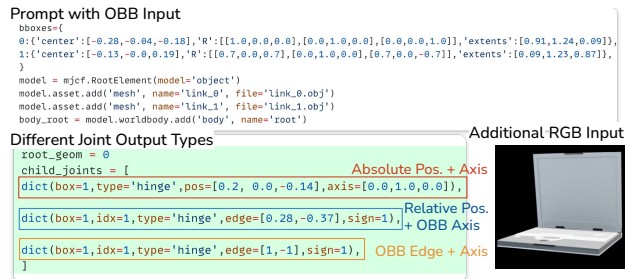

Figure 5: Qualitative comparison of the code output format in ablation experiments. Each formulation occupies one line. In 'Absolute Pos. + Axis', LLM outputs continuous position and axis values; in 'Relative Pos. + OBB Axis', LLM outputs one index into the OBB's rotation axis, and a 2D joint position relative to the selected axis; Real2Code uses 'OBB Edge + Axis', where LLM outputs index to rotation axes in an OBB, and two values to indicate the OBB edge. Bottom right of the figure shows one example of additional RGB image input to the LLM.

## 5 CONCLUSION

We present Real2Code, a novel method for reconstructing articulated objects that leverages the capability of pre-trained vision and large language models. We empirically show that Real2Code achieves a new state-of-the-art in both geometry reconstruction and articulation prediction. We hope Real2Code unlocks new opportunities in robotics and mixed reality applications.

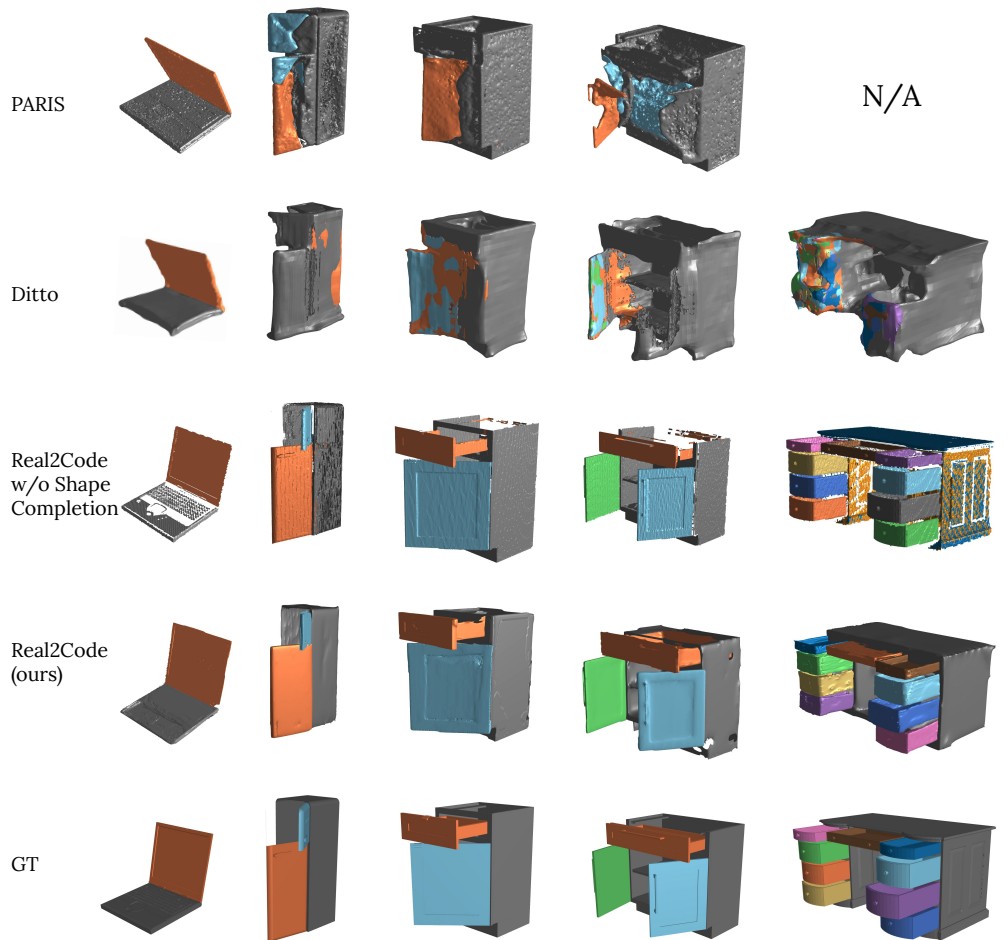

Figure 6: Qualitative results that compare Real2Code to baseline methods. We show test on objects with varying kinematic complexities, from a two-part laptop to a ten-part multi-drawer table. Whereas all methods handle the simpler laptop articulation, baseline methods struggle as the number of object parts increases, and Real2Code performs reconstruction much more accurately. PARIS runs out of memory and fails to run on the ten-part table object ('N/A').

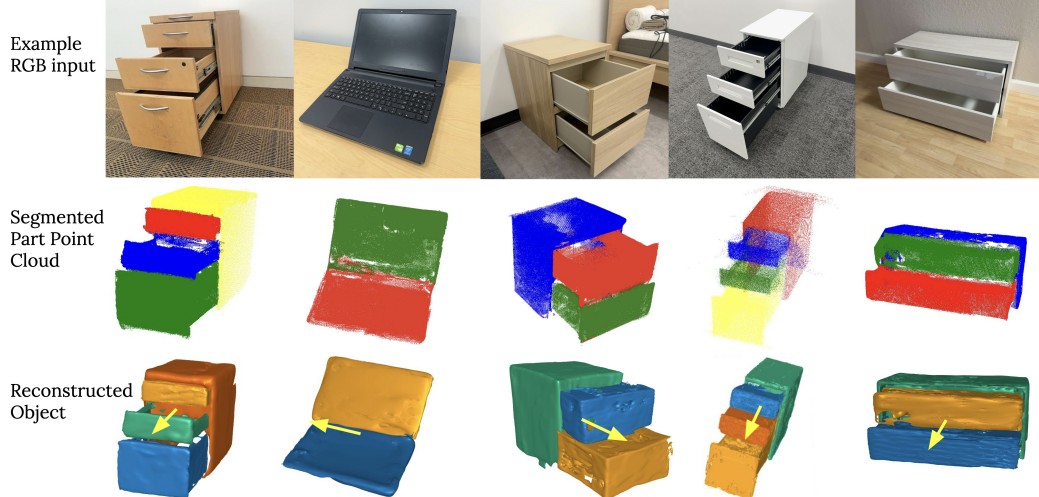

Figure 7: We evaluate Real2Code on real world objects using RGB data. For each object, we use 10 pose-free RGB images captured in-the-wild and run Real2Code with DUSt3R(Wang et al., 2023b). We show one example RGB input (1st row), segmented point clouds (2nd row) and full reconstruction (3rd row) for each object.

# 6 ACKNOWLEDGMENTS

This work was supported in part by the Toyota Research Institute, NSF Award #2143601, Sloan Fellowship. The views and conclusions contained herein are those of the authors and should not be interpreted as necessarily representing the official policies, either expressed or implied, of the sponsors. The authors would like to thank the following friends and colleagues for help during various stages of the project: Zhenjia Xu, Samir Gadre, Cheng Chi, Hanxiao Jiang, Bingjie Tang, and members of REALab: Zeyi Liu, Xiaomeng Xu, Chuer Pan, Huy Ha, Yihuai Gao, Mengda Xu, Austin Patel for valuable feedback and discussion on the paper manuscript. We also thank administrators of Stanford EE department, namely Kenny Green, Steve B. Cousins and Mary K. McMahon for their support during real world data collection.

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

# A APPENDIX

## A.1 DISCUSSIONS & LIMITATIONS

In this section, we discuss a few key limitations that point to interesting directions for future work:

1. Real2Code currently handles one single object at a time. To achieve scene-level reconstruction, i.e. multiple objects each with multiple articulated parts, additional processing is required on top of the current pipeline. For example, given a sequence of multi-view image inputs of a multi-object scene, we can first use an object detection model to single-out each detected object, then use the original Real2Code pipeline to handle object-level reconstruction.

2. Test-time computational efficiency. Due to the iterative prompting method described in 3.1.2, i.e. 1) grid-sampling of prompt points on each single RGB image and 2) prompting on all camera views for each object part, our test-time compute for SAM forward pass scales linearly with the number of input camera views: if an object has $M$ camera views, $K$ object parts, and uses $NxN$ grid for initial prompt points sampling (we use $N = 16$), then the SAM Kirillov et al. (2023) is prompted with $N + K(M - 1)$ single 2D-point and RGB image pairs. Notably, $M$ is the main compute bottleneck because we can cache the image embedding from SAM Kirillov et al. (2023), and only call the light-weight prompt decoder for additional prompt points. Additionally, the inference compute for LLM code generation is dependent on the number of object parts, and roughly scales linearly with the generation token length. Overall, our system is slower at inference time when compared with end-to-end methods such as CARTOHeppert et al. (2023) and Ditto Jiang et al. (2022b), but is more scalable to more complex articulation structures because it handles arbitrary numbers of object parts.

3. Cascading dependency. Because Real2Code is composed of multiple modules, failure cases happen when the errors from each module propagate and lead to sub-optimal final object reconstructions. We found that the articulation prediction accuracy is sensitive to failures in the first 2D image segmentation module, i.e., OBBs from wrong segmentations directly obstruct the LLM reasoning of object structures. To increase robustness, we can improve the system by providing human corrective feedback as proposed in Kirillov et al. (2023), i.e., a user provides additional points and prompt the model to adjust its mask predictions. Then only feed the input with satisfactory OBB extractions to LLM for code generation.

4. Objects with hinge joints that do not overlap with OBB edge. To handle new object categories, we remark that 1) the geometry reconstruction part of Real2Code (both part segmentation shape completion) can handle complex geometry shapes (e.g. scissors, faucet handles) when given the training data for fine-tuning the part segmentation model and shape completion model. 2) However, because we select OBB edge as rotation center, our method can handle sliding joints (e.g. a sliding oven rack) but will be inaccurate for hinge joints where the joint is not overlapping with any OBB edge (e.g. scissors). To address this, one could add another fine-tuning head to further adjust the LLM outputs (which selects one OBB edge) by predicting an offset value to improve the joint position accuracy.

## A.2 DATASET PREPARATION DETAILS

**Base: PartNet-Mobility Object Assets.** We use the same set of 467 training and 35 testing objects from 4 categories in PartNet-Mobility (Mo et al., 2019). The raw dataset contains a rich collection of object meshes, textures, and URDF files that contain articulation information. We further process the data as follows:

**RGB-D Image Rendering** We render each object individually using Blender (Community, 2018; Denninger et al., 2023) for 5 loops. For each rendering loop, the object is centered at the scene origin and the rendering camera poses are randomly sampled; we render 12 RGB-D images and all the segmentation masks corresponding to the all the object parts. During rendering, we also randomly sample joint states in the object such that all its doors or drawers are partially open — we make the assumption that all the parts our train and test objects are partially open to remove ambiguity and provide more observation view into object insides.

**Mesh Pre-processing.** The original PartNet-Mobility assets contain highly fine-grained meshes, i.e.,one drawer part is comprised of more than ten panel or bar-shaped meshes. To prepare data for part-level shape completion, we group these fine-grained meshes such that meshes from the same object part are merged into one single mesh. Mesh textures are ignored during grouping, resulting in grouped texture-less part-level meshes. The RGB-D images and masks are then used to generate part-level point clouds as partial observations. We use Kaolin (Fuji Tsang et al., 2022) to sample label occupancy values from object part meshes.

**Code-Generation Data.** To prepare data for fine-tuning code-generation LLMs, we first use the rendered RGB-D images and segmentation masks to obtain *ground-truth* part-level point-clouds, which are used to extract oriented-bounding boxes (OBBs) for each part. Next, we take the raw object URDF files and generate a shorter

copy with our grouped part meshes. Because the raw URDF/XML syntax contain long unnecessary details, we manually translate them into Python-like MJCF (Tunyasuvunakool et al., 2020) code, which are a lot more compact and familiar to the pre-trained LLMs. Finally, for each of the 5 rendering loops per object, we re-write the object code again to replace the absolute joint information with the relative position and rotation of each joint with respect to the extracted OBBs. We further augment the data by randomly rotating the OBBs along the z-axis, 5 times per object. This results in $468 \times 5 \times 5 = 11700$ training samples for LLM fine-tuning.

## A.3 MODEL TRAINING DETAILS

**SAM Fine-tuning.** The fine-tuning data consists of $28,020$ RGB images, and each image corresponds to a set of binary segmentation masks, one per each object part plus a background mask. We fine-tune only the decoder layers of pre-trained SAM (Kirillov et al., 2023) on this custom dataset while keeping the rest of the model weights frozen. Each fine-tuning batch contains 24 RGB images; for every RGB image in the batch, we sample 16 prompt points uniformly across each image's ground-truth masks, i.e.,only sample points from the positive mask area. Hence each training batch of size $B = 24$ contains 24 images and $24 \times 16$ pairs of prompt point and ground-truth masks. Following the original paper (Kirillov et al., 2023), we update the model with a weighted average of Focal Loss (Lin et al., 2018), Dice Loss (Sudre et al., 2017) and MSE IoU prediction loss.

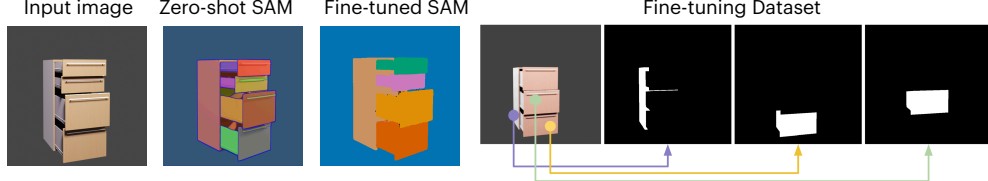

Figure 8: **Kinematics-aware SAM Fine-tuning.** Given an RGB input image, the pre-trained zero-shot SAMKirillov et al. (2023) produces unnecessarily detailed segmentation masks (column Zero-shot SAM'. We construct a dataset of objects' RGB images and kinematics-aligned ground-truth masks (column 'Fine-tuning Dataset'). The model is fine-tuned to take one image and one sampled 2D query point and predict the corresponding part mask. We compare the output of the model after fine-tuning on the same image (column 'Fine-tuned SAM').

**Training Shape Completion Model.** We use $6,260$ pairs of partial point clouds and size $96^3$ occupancy grids and train our PointNet++ (Qi et al., 2017) based occupancy prediction model from scratch. For a training batch of size $B$, we sample $B$ point clouds of size 2048, and sample $B \times 12,000$ query points on the label occupancy grids. Notably, because object parts are of different scales, we normalize the occupancy grid using *partial* OBBs extracted from the input point cloud to avoid under-fitting the smaller-sized meshes. When sampling training query points, we found sampling 25% occupied works the best for balancing between occupied areas and empty space, and we add a random shifting step on the occupied grids to improve model accuracy on the near-surface areas. At test time, we query on a $96^3$ grid and use Marching Cubes (Lorensen & Cline, 1987) to extract the completed part meshes.

**Fine-tuning Code Generation LLM.** We use the pre-trained Codellama (Rozière et al., 2023)-7B model on our code dataset, which contains code samples generated from PartNet (Mo et al., 2019) objects as described above. We use LoRA (Hu et al., 2021), a low-rank weight fine-tuning technique, to fine-tune the model with the next-token prediction loss. For training efficiency, we compress the training sequences by removing unnecessary empty character spaces and overhead code lines (such as package import statements). The resulting training set contains under 800 tokens per sequence for objects with up to 7 parts (i.e., 6 articulated joints). Despite the short training data, we found the model to be able to extrapolate to unseen test set objects with up to 15 parts when the ground-truth segmentation is provided.

## A.4 DETAILS ON REAL WORLD EVALUATIONS

**Data Collection.** We collect data from a set of common furniture objects, including cabinets, laptops, night stands, dressers, ranging from 1 to 3 moving parts. Each object is scanned using a LiDAR-equipped iPhone camera and 3dScanner App (3D Scanner App, 2024) to capture a series of RGB images from the front $180°$ view. We then select 10 RGB images per object, and crop and resize them into $512 \times 512$ images used by SAM (Kirillov et al., 2023) and DUSt3R (Wang et al., 2023b).

**Part Segmentation from Unstructured RGB images.** Fig. 9 visualizes the DUSt3R model output on an example object in: notably, the model predicts dense point-maps on the object's surface area that can be globally-aligned into a object point cloud; but the 3D points are less accurate on the partially occluded areas,

Globally Aligned Dense Point-maps from DUSt3R      View-consistent Prompt Points

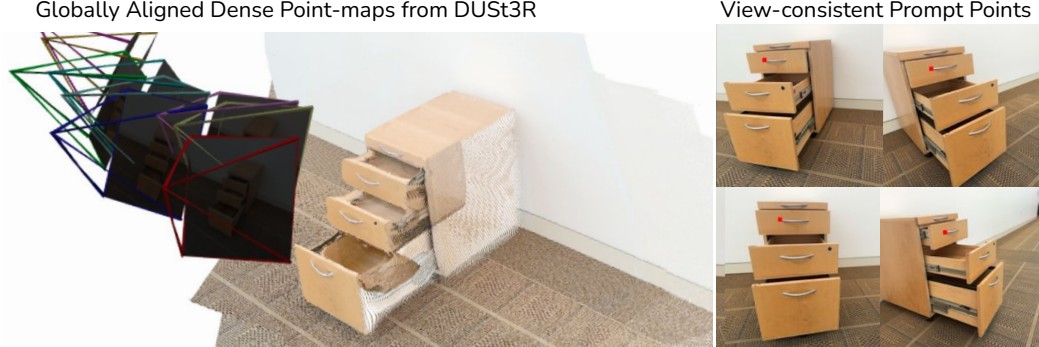

Figure 9: **3D part segmentation from Pose-free RGB images.** Illustration of how DUSt3R (Wang et al., 2023b) is used to achieve 3D part segmentation from unstructured RGB images. For each object, we take around 10 pose-free RGB images as input to the pre-trained DUSt3R (Wang et al., 2023b) model, which outputs a set of globally-aligned 2D-to-3D dense point-maps, i.e.,every 2D pixel on each image is matched to a point in 3D. This correspondence enables cross-view pixel matching via finding nearest-neighbor in 3D space. We can therefore sample view-consistent 2D points for prompting our fine-tuned SAM model, and the resulting segmented masks are grouped into 3D part segmentation.

Jewelry Box                      Gameboy

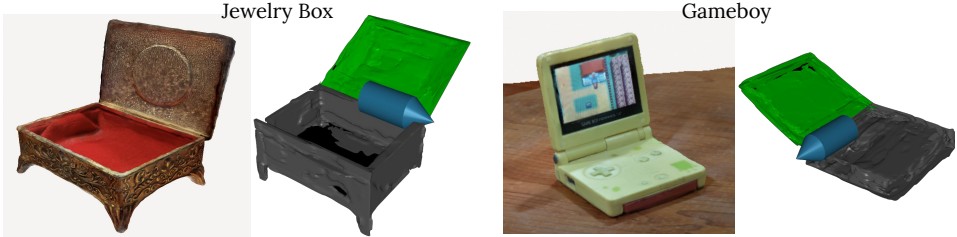

Figure 10: We demonstrate that Real2Code can be used for labeling and animating real world objects. We evaluate Real2Code on scanned real objects from Polycam (Polycam, 2024) and export the resulting mesh and joints in MuJoCo (Todorov et al., 2012). Blue arrows indicate the simulated joint axis and position; mesh corresponding to the moving part is colored in green.

such as the inside of the drawer. This is likely due to these areas are less common in the model's pre-training dataset. Also notice that, because we sample each 2D point from one RGB image first and uses nearest neighbor in the predicted 3D point-map to find its matching 2D point in another image, it might find a wrong match if the point is occluded and not visible in the other image. We address this by manually setting a distance threshold, and decide a match cannot be found if its 3D point's distance to the nearest neighbor is above set threshold.

## A.5  ADDITIONAL RESULTS ON ANIMATING SCANNED REAL WORLD OBJECTS

In addition to object reconstruction from raw RGB images, we show Real2Code can also be used to animate scanned objects. We use real world scanned object meshes uploaded by users of the Polycam (Polycam, 2024) App, and use our Blender rendering pipeline to render RGB-D images. We evaluate our image segmentation, shape completion, and code generation models on these images, and demonstrate only the qualitative results due to the lack of ground-truth data. We execute the final model output code to show the objects can be simulated in MuJoCo (Todorov et al., 2012). See Fig. 10 for visualizations. These real world objects feature complex visual appearance that falls outside our SAM fine-tuning distribution, but Real2Code is still able to successfully segment parts and predict reasonable joint positions and rotations.

## A.6  ADDITIONAL BASELINE RESULTS

| Type | ID | Art. Parts | CD (whole)↓ | CD (part)↓ | rot↓ | pos ↓ |
|------|-----|-----------|-------------|------------|------|-------|
| Refrigerator | 12043 | 2 | 1.01 | 0.594 | 0.19 | 0.05 |
| Refrigerator | 12059 | 2 | 0.84 | 0.504 | 0.24 | 0.014 |
| Refrigerator | 12066 | 2 | 0.78 | 120.32 | 8.86 | 2.07 |
| StorageFurniture | 45694 | 2 | 0.98 | 0.41 | 0.06 | 0.0 |
| StorageFurniture | 44853 | 3 | 1.71 | 101.09 | 37.84 | 0.0 |

Table 4: Additional experimental results that evaluates DigitalTwinArt Weng et al. (2024) (a concurrent work) on objects with 2 or more moving parts. We evaluate each object on 3 individual seeds and report the average performance across 3 runs. We show that, although this method reported better performance than our compared baseline method PARISLiu et al. (2023a), it still struggles with objects with more than 1 moving parts and under-performs Real2Code.

