# OpenReview forum: "Real2Code: Reconstruct Articulated Objects via Code Generation"
_ICLR.cc/2025/Conference — ICLR 2025 Poster_

### Official Review · Reviewer_mGLU · 2024-10-27

**Soundness:** 3
**Presentation:** 3
**Contribution:** 3
**Rating:** 6
**Confidence:** 4

**Summary:**

This paper presents Real2Code, a novel approach for reconstructing articulated objects from visual observations.
Main Contributions:
1. A new method (Real2Code) that reconstructs articulated objects by generating code, using fine-tuned LLMs specialized for this task.
2. A part reconstruction pipeline that combines:
    - Kinematic-aware view-consistent part segmentation model.
    - 3D shape completion model.
    - Fine-tuned LLMs to predict joint articulation.
3. Significant performance improvements over previous methods:
    - First approach to accurately handle objects with more than three parts
    - Generalizes beyond training data (trained on up to 7 parts, works on up to 10 parts)
    - Works with just a few RGB images, without requiring depth or camera information

**Strengths:**

1. The writing is clear and easy to follow.
2. The proposed pipeline innovatively formulates the articulation reconstruction as code generation, which naturally combine current powerful foundation models (SAM, LLM) for articulated object reconstruction.
3. Real2Code demonstrates significant performance improvements over previous methods. It accurately handle objects with more than three parts and only requires RGB images without requiring depth or camera information.
4. This paper provides details for the training of the whole pipeline, including data preparation and training of key components (SAM, completion model, and CodeLlama), which demonstrates good reproducibility and technical soundness.

**Weaknesses:**

1. Real2Code demonstrates good performance on trained categories (Laptop, Box, Refrigerator, Storage-Furniture, and Table). However, the performance of unseen categories is not explored. I don't expect the model to generalize well to all other categories, but I do expect some experiments to show whether there is still a problem with category generalization.
2. There is no discussion of when the model will fail, especially if some of the components of the model fail.  For example, fine-tuned SAM might not segment parts accurately, or the LLM might output an incorrect result under certain circumstances.


These weaknesses don't necessarily diminish the paper's contribution but addressing them would strengthen the work and increase its impact. Many could be addressed through additional experiments and analysis rather than fundamental changes to the method.

**Questions:**

1. Why is the method from [1] not included in the baseline comparisons, given that it demonstrates better performance than PARIS?
2. How is the cross-category generalization ability of Real2Code, especially on real-world data?
3. How long do the training and inference of the entire pipeline take respectively？
4. How to extract oriented bounding box of each part?

[1] Yijia Weng, Bowen Wen, etc. Neural implicit representation for building digital twins of unknown articulated objects.

---

> ### Comment · Reviewer_mGLU · 2024-12-02
>
> The authors have not made any response to my concerns and questions as well as those of other reviewers. Considering my concerns and those of other reviewers, I reduce the score to 6.

---

> ### Author Response · Authors · 2024-12-02
> **updated full response**
>
> Dear reviewer#mGLU,
>
> Apologies for the late response, we were still in progress of gathering experimental results to better respond to your comments (e.g. running additional experiments on novel category objects, evaluating the cited method [1], which were time-consuming). We are updating with a better-formatted full response here. We hope you could consider changing back the score.
>
> _Weaknesses section_
>
> **W1 Performance on new or unseen categories**
>
> We provide additional experiment results, which are divided into two sets
> 1. Evaluation on smaller objects with complex shapes. We prepared data for two new object categories from PartNet-Mobility - Eyeglasses and Scissors. Both categories have complex shapes beyond simpler, cuboid-like shapes. We use the new dataset to re-train both our proposed SAM fine-tuning model and shape-completion model as described in the main paper, and test the new models on 8 unseen, held-out instances.
>
> Please see this link: https://sites.google.com/view/real2code-submission/complex-shapes for more details and result visualizations. These objects post additional challenges due to having parts with thinner shapes (e.g. blades) and containing more intricate details (e.g. handlebars), and taking up smaller areas in the rendered RGBD images. Qualitatively, we observe the SAM-based segmentation model is able to propose kinematically correct segmentations, but fails at some instances where the glasses legs or scissor blades are too thin.
>
> 2. Generalization to novel object categories.
>
> We additionally added 8 more test objects from Microwave and Door categories of PartNet. These two categories were never seen in our training set. Many Microwave or Door objects have similar structural complexity to the StorageFurniture objects we trained on (e.g. a microwave also has a box-like parent body and a hinge door), but contains novel visual appearances and new geometries unseen from the training set. Therefore, our LLM module can generalize to these objects, but our fine-tuned SAM model and our shape completion model does not handle the OOD geometries well (e.g. the press buttons in a microwave dial panel). We have provided the additional details and results on these generalization experiments at link: https://sites.google.com/view/real2code-submission/generalization-to-novel-category
>
>
> **W2 regarding error propagation**
>
> This is indeed a limitation of our method. Overall, the main bottleneck of our system is the part-level segmentation quality, because as long as a valid part gets segmented out into a reasonable point-cloud, 1) the shape completion model is trained with noisy input such that it can complete the input; 2) it will result in a reasonable OBB as input to the LLM, which is essentially copying over one of the input edges as a joint axis. In the updated submission pdf, we have added a Discussion & Limitation section to ensure this limitation is well disclosed.
>
> To further improve generalization, an interesting future direction would be involving SAMv2 to obtain more robust segmentations, or using large-scale pre-trained 3D generative models to complete the object shape. See: https://sites.google.com/view/real2code-submission/dataset-details for an illustrated demo of incorporating SAMv2
>
>
> _Questions section_
>
> **Q1 regarding comparison with [1]**:
>
> Due to limited space, please see the other comment below for more details.
>
> **Q2 on cross-category generalization**:
>
> Following the discussion above and results here: https://sites.google.com/view/real2code-submission/generalization-to-novel-category  We observe that, again, 2D part segmentation is the main bottleneck of our pipeline (our model's mesh completion from GT segmentation has clearly better quality), and completely fails for more extreme OOD instances where the object looks very different from training set (e.g. the glass door ID 9107).
>
> **Q3 on training & inference time**:
>
> Training time for SAM fine-tuning, shape completion model training, and LLM fine-tuning takes approximately 24hrs, 12hrs, and 10hrs respectively. At inference time: 1) SAM prompting takes up approximately 3min per object because of our view-consistent prompting scheme, but the inference time is still kept reasonable because we cache the image embedding from SAM and only call the light-weight prompt decoder for later prompt points; 2) shape-completion model forwarding is single-pass, which can be batched and takes <1min for all parts; 3) LLM generation takes ~2min per object, because the output is strictly formatted to be concise, it requires only ~200 output token length.
>
> **Q4 on extracting oriented bounding box**: we first use the multi-view RGBD images or the Dust3r-output point maps to get segmented part-level point clouds, then extract OBB using open3d (`open3d.geometry.OrientedBoundingBox`).
>
>
> Thank you for the detailed reviews and feedback. and please let us know if you have further questions.

---

> > ### Author Response · Authors · 2024-12-02
> > **Additional results from evaluating DigitalTwinArt [1]**
> >
> > DigitalTwinArt [1] is a concurrent work which we did not have access to evaluation code during the time of submission. Due to limited time, we selected five multi-part objects with 2 or more moving parts for evaluation. To provide a fair evaluation as we did for PARIS baseline, we evaluate each object on 3 different seeds (i.e. a separate optimization run per seed), and report the average performance across 3 runs. Please see the updated submission website for the result table containing averaged shape reconstruction and joint prediction results, as well as visualizations of the evaluated objects. https://sites.google.com/view/real2code-submission
> >
> >
> > Albeit not a full comparison with all the test objects reported in the main paper, these results should provide sufficient evidence that although DigitalTwinArt reported better performance than our compared baseline method PARIS and is significantly more stable to optimize (i.e. results from different runs show a lower variance), it still struggles with objects with more than 1 moving parts and overall under-performs Real2Code.
> >
> > [1] Yijia Weng, Bowen Wen, etc. Neural implicit representation for building digital twins of unknown articulated objects.

---

### Official Review · Reviewer_1whT · 2024-11-01

**Soundness:** 3
**Presentation:** 3
**Contribution:** 2
**Rating:** 6
**Confidence:** 3

**Summary:**

This paper formulates joint prediction as a code-generation problem and adapts LLM to this task, which makes it scale elegantly to process an articulated object with multiple joints. It also introduces a part reconstruction pipeline leveraging 2D part segmentation and part-level shape completion.

**Strengths:**

- Formulating joint prediction as a code-generation problem provides an elegant way to handle varying numbers of object joints.
- Part-level shape completion makes sense since part structures are much simpler than structures of whole objects. Table 1 demonstrates the effectiveness of the proposed shape completion model.

**Weaknesses:**

The selection of object categories for evaluation is limited.
- For part-level shape completion, it would be more compelling to include categories with a greater diversity of part shapes rather than focusing primarily on cuboid-like forms. For instance, objects such as globes and lamps in PartNet Mobility exhibit a variety of shapes, including spherical and cylindrical forms, which provide a more comprehensive basis for evaluation. Additionally, the assumption that 'many common articulated objects consist of cuboid-like parts' is not fully substantiated when considering the full range of object categories in PartNet-Mobility.
- In articulation prediction, the formulation assumes that 'the position of corresponding revolute joints will lie closely to, if not overlap with, one of the OBB edges'. However, this assumption seems not to be solid enough either. Take “folding chairs” in PartNet-Mobility for example, the revolute joints of many instances lie not close enough to OBB edges (quadrisection point or even trisection point). Do these assumptions restrict the range of categories suitable for evaluation?

**Questions:**

Why were only these five object categories chosen from PartNet-Mobility for evaluation? The current formulation relies on assumptions that appear somewhat unsubstantiated. Is this why Real2Code is hard to evaluate in more diverse categories?

---

> ### Author Response · Authors · 2024-12-03
>
> Thank you for the detailed reviews and feedback. We hope our additional clarifications, experiments, and the updated submission pdf will address your raised concerns and increase your confidence in accepting our submission, and please let us know if you have any further questions.
>
> **Assumption on object joint structure**
>
> Our pipeline can be divided into two parts: 1) geometry reconstruction (includes 1.1 part segmentation and 1.2 shape completion); 2) articulation prediction. We’d like to clarify that our method for 1) can handle arbitrary geometry shapes (e.g. scissors, faucet handles), if these objects are put into the training set for fine-tuning the part segmentation model and shape completion model. However, for 2), because we use the OBB formulation and select OBB edge as rotation center, our method can handle sliding joints (e.g. a sliding oven rack) but will be inaccurate for hinge joints where the joint is not overlapping with any OBB edge (e.g. scissors). To also handle these objects, one possible extension is to add a regression MLP that takes in the OBB information and the LLM’s selection of rotation direction, then predicts a more precise joint axis. We have updated the submission pdf to better clarify our assumption and the subsequent limitations. Since cuboid-shaped objects (cabinets, doors) are commonly seen, we still believe it’s of much value to reconstruct those objects, especially those with many parts, which our method handles much better.
>
>
>
>
> **Evaluation on more object categories**
>
> Following the discussion above, we provide additional experiments to further validate (1), i.e. and to test whether Real2Code can handle objects with complex geometries. We collected a new dataset that included two new object categories from PartNet-Mobility - Eyeglasses and Scissors. These two categories have complex shapes beyond simpler, cuboid-like shapes (e.g. cabinets) that we evaluated on in the main paper. To prepare the dataset, the original PartNet object assets require mesh repairing processing, manual cleanup, deduplication (several instances are repeated and got removed to keep the train-test split clean), and render RGBD images and occupancy grids. This results in 52 and 43 training objects for Eyeglasses and Scissors, respectively, a total of 20200 segmentation masks and 14520 part ground-truth meshes.
>
> We use the new data to re-train both our proposed SAM fine-tuning model and shape-completion model as described in the main paper, and test the new models on 4 unseen, held-out object instances from each category. Please see more details and qualitative results here: https://sites.google.com/view/real2code-submission/complex-shapes. These objects are challenging due to having parts with thinner shapes (e.g. blades) and containing more intricate details (e.g. handlebars),  and taking up smaller areas in the rendered RGBD images.  Qualitatively, we observe the SAM-based segmentation model is able to propose kinematically correct segmentations, but fails at some instances where the glasses legs or scissor blades are too thin.
>
> We have not added more categories for now due to time constraints in data processing and model training (for the Lamp and Globe objects mentioned by the reviewer, the mesh processing for the assets are quite demanding), but hope these results provide sufficient evidence that our method can handle more diverse and complex object geometry.

---

> > ### Comment · Reviewer_1whT · 2024-12-03
> >
> > Thank you for the response and the additional qualitative results. They have addressed my concerns, and I am raising my score to 6.

---

> > > ### Author Response · Authors · 2024-12-03
> > >
> > > Thank you for revisiting the score. We appreciate your time dedicated to reviewing our submission and your feedback that helps us improving this work!

---

### Official Review · Reviewer_GDby · 2024-11-04

**Soundness:** 3
**Presentation:** 3
**Contribution:** 3
**Rating:** 6
**Confidence:** 3

**Summary:**

The paper focuses on the task of articulated objects reconstruction given only a few images of an object. The pipeline proposed first reconstructs parts from images and then leverages LLM to predict the joint parameters, which generalizes to objects with multiple joints. The method is evaluated on five categories in the PartNet-Mobility dataset and outperforms previous methods.

**Strengths:**

1. The paper proposes a new pipeline Real2Code to reconstruct articulated shapes from images. It shows promising results on multiple categories with different joint types in the PartNet-Mobility dataset.

2. I find its generalization ability to multiple-joint shapes particularly interesting, which could potentially enable many real-world robot manipulation tasks.

3. The paper is overall easy to read.

**Weaknesses:**

1. The proposed pipeline consists of multiple components and, as a result, rather fragile from what I understand, since a failure in any component in the middle can cause the entire pipeline to break down. For example, if the part bounding boxes parameters (segmentation or shape completion) are inaccurate, the joint prediction part will carry these errors. Since the whole procedure is open-loop, I wonder if the method still produces reasonable shapes assuming initial bounding box predictions are inaccurate?

2. The method is only evaluated on five categories, and these categories (Box, Refrigerator, Storage-Furniture and Table) are all quite similar in topology, similar for the real-world examples. So I think it would be helpful to see results on more diverse shapes. In addition, is CodeLlama trained on all categories together? How does CodeLlama handle scale differences of different objects / categories? Or all shapes normalized so the input to the LLM is kind of already normalized?

3. Which component is the bottleneck of the pipeline? Is it the part segmentation or joint prediction of CodeLlama? Ablation studies on this are essential to better evaluate the approach.

4. To better support the claim of being able to extrapolate beyond objects’ structural complexity in the training set, I think it would be important to provide more results. For example, does the trained model generalize to other categories?

**Questions:**

The results presented in the paper are interesting, but I believe that additional evaluations would strengthen the significance and impact of the work.

---

> ### Author Response · Authors · 2024-12-03
>
> Thank you for the detailed reviews and feedback. We hope our additional clarification, experiments, and the updated submission pdf will address your raised concerns and increase your confidence in accepting our submission, and please let us know if you have any further questions.
>
> **Question 1 & 3: Fragile to failures in specific components in the pipeline, and system bottleneck**
>
> - Error propagation is a valid limitation of our method. Overall, the main bottleneck of our system is the part-level segmentation quality, because as long as a valid part gets segmented out into a reasonable point-cloud, 1) the shape completion model is trained with noisy input such that it can complete the input; 2) it will result in a reasonable OBB as input to the LLM which is essentially copying over one of the input edges as a joint axis.
>
> - This can be better illustration by the qualitative results in here: https://sites.google.com/view/real2code-submission/complex-shapes We observe when give ground-truth part-level segmentations, the shape completion model produces high quality mesh reconstructions, and the resulting OBBs are naturally well fitted and oriented. However, when there are sufficient errors in some of the segmented views, we see the segmented point clouds having “leakage” between object parts. See Eyeglasses object#101303: The resulting mesh is inflated because the input point cloud on the glasses part contains inaccurate points on the legs, hence the model receives a bad normalized input pcd (recall that we normalize all the part point clouds using partially-observed OBB during inference).
>
> - In the updated submission pdf, we have added a Discussion & Limitation section and ensure this limitation is well disclosed (see Appendix section A.1.3). And we will update the submission with these additional results to provide more insights. However, there is also several possibilities to increase the robustness of our part segmentation module: 1)  provide additional user prompt points, this will allow the model to focus more on the correct parts; 2) incorporate the latest SAMv2 model that segments videos. We can concatenate multi-view inputs into a video, first run our fine-tuned model to propose kinematically-accurate parts, then run SAMv2 to obtain view-consistent masks. We provide a demo of this process in the video here: https://sites.google.com/view/real2code-submission/dataset-details
>
>
> **Q2: Results on more object categories.**
> - We have added additional experiments to evaluate our method on new object categories, please see qualitative results here: https://sites.google.com/view/real2code-submission/complex-shapes
> We prepared a new dataset using two new object categories from PartNet-Mobility - Eyeglasses and Scissors, both have complex shapes beyond simpler, cuboid-like shapes (e.g. cabinets) that we evaluated on in the main paper. We use the new data to re-train both our proposed SAM fine-tuning model and shape-completion model as described in the main paper, and test the new models on unseen, held-out object instances. Qualitatively, we observe the SAM-based segmentation model is able to propose kinematically correct segmentations, but fails at some instances where the glasses legs or scissor blades are too thin.
>
> - Regarding questions on LLM training details: 1) yes, LLM trained on all categories together. 2) the objects across categories are normalized to the same scale in size, subsequently CodeLlama inputs are also normalized. See https://sites.google.com/view/real2code-submission/dataset-details for a mesh visualization of a Laptop object and a Box object, which we show are already on a matching scale.
>
>
> **Q4: Generalization to other categories:**
> - Take the example of Microwave objects from PartNet. Microwave is an unseen category not included in our training set, objects here have similar structural complexity to the StorageFurniture objects we trained on (a box-like parent body and a hinge door), but contains novel visual appearances and new geometries unseen from the training set. Therefore, our LLM module can generalize to these objects, our fine-tuned SAM model sometimes fails on object instances that have OOD visuals, and our shape completion model does not handle the geometries well (e.g. the press buttons in the dial panel).
>
> - Overall, because of our OBB formulation, the LLM can generalize reasonably within similar structures to our training set. In terms of geometry reconstruction, the generalization is bounded by both the visual appearances and diversity in shape completion model training data. To further improve generalization, an interesting future direction would be involving SAMv2 to obtain more robust segmentations, or using large-scale pre-trained 3D generative models to complete the object shape. We have provided the additional details and results on these generalization experiments at link: https://sites.google.com/view/real2code-submission/generalization-to-novel-category

---

### Official Review · Reviewer_9fCy · 2024-11-04

**Soundness:** 2
**Presentation:** 2
**Contribution:** 2
**Rating:** 6
**Confidence:** 4

**Summary:**

This paper introduces Real2Code, a method for reconstructing articulated objects from multi-view images through code generation. The method first reconstructs part geometry using image segmentation and shape completion. Then it predicts joint information as code generation from fine-tuned LLM which takes an object part as oriented bounding boxes. Experiments show that this method outperforms previous method in generating parts with over three parts and can generalize to real object reconstruction by training only on synthetic data.

**Strengths:**

1. This method formulates joint prediction as a code generation problem, which is different from prior work. The biggest advantage of such a formulation is the ability to scale well with different numbers of parts (prior work works mostly for objects with <=3 parts).

2. The overall pipeline is novel -- it leverages a few different modules including Vision models for part segmentation and completion as well as LLM for code generation. This way, the problem is decomposed into a few smaller steps which is shown solvable with previous methods.

3. The text and figures are overall well-written and easy to follow.

4. Experiments have been conducted to validate each proposed components. Results seem to achieve state of the art, especially on objects with many parts.

**Weaknesses:**

1. In Sec. 4.2.1, it mentions that "we generate permutations of the set of predicted meshes and take the permutation that results in lowest error; the same logic is used for joint prediction results". I was wondering why this is needed to evaluate this method. Is it because the proposed method is not very stable? How much more time would this cost for the inference of this method?

2. The link to more visualizations included in Sec. 4.4 does not contain any result visualizations -- it seems it only has a method overview figure and an abstract.

3. The content in Tab. 1 is a bit confusing:
(1) what is ``Real2Code+gtSeg``, the paper does not seem to mention / analyze this row anywhere in the text.

(2) If I understand ``Real2Code+gtSeg`` the same way as ``Real2Code+gtBB`` in Tab. 2, it should be an upper bound of ``Real2Code (Ours)``, if so, why does ``Real2Code+gtSeg`` perform worse than ``Real2Code (Ours)`` in a few columns like Whole & Parts for Box, etc.

**Questions:**

1. Tab. 3 and its corresponding text has some typos: row 2 has "Rot" in out column, but it is referred to as "Rel" in the text (if I understand it correctly).

2. In Tab. 3, the first row has 0 error for "rot" on 3, 4-5, 6-15 parts. Then why the rot error suddenly becomes very big for 2 parts?

3. How do you determine the parent vs. child node / the canonical pose, especially for real-world objects? For example, the two parts of a laptop have very similar geometries/OBB. If a laptop is placed upside down, would this method also instead treats the keyboard part is child and the screen part as parent?

---

> ### Author Response · Authors · 2024-12-02
>
> Dear Reviewer 9fCy
>
> Thank you for the detailed reviews and feedback. We hope our additional clarifications and the updated submission pdf will address your raised concerns and increase your confidence in accepting our submission.
>
> **Weaknesses section**
> 1. Q: Reason for using permutations during evaluation.
>
> A: This is because our method does not get explicit information about how many parts to predict for a given test object or which specific part to reconstruct,  i.e. it must predict all parts together, and it may or may not match the ground truth. Therefore we use permutation as an automatic way to evaluate all possible matches and take the lowest error one. We believe this is a more systematic way of evaluating the predictions, and not inherently an issue with our method or problem formulation.
>
> 2. Q: links to visualizations
>
> The animated figure was intended to show that our reconstructed object can be directly imported to a physics simulation (MuJoCo) and have a robot arm manipulating it. We have also since updated the submission website to include more visualizations and additional results requested during the rebuttal phase.
>
> 3. Q: confusing content in Tab. 1
>
> A: Apologies for the confusion. The  Real2Code+gtSeg result is to validate the need for our shape completion model: we use ground-truth image segmentation to get correct but partially observed point clouds (because the object observations contain occlusions). This subsequent mesh extraction from the aggregated point clouds is better than using SAM segmentation, but is still incomplete and hence leads to larger chamfer-distance errors.  We have also updated section 4.2.1 to better clarify the meaning of Real2Code+gtSeg
>
> **Questions section**:
> 1. Q: Typo in Table 3.
>
> A: We have updated Table 3, Cls. should indeed be Rel.
>
> 2. Q: Reason for larger error in Tab.3 “OBB + Rot”
>
> A: Because our error is measured in degrees, getting one joint wrong will lift the average error from 0.0 to 5.0-degree (because error angle is 90-degree and we evaluate on 9 test objects with 2 joints each.
>
> 3. Q: how to determine parent/child
>
> A: The model first predicts a parent body (this is the ‘root_geom’ prediction in Figure 4), and every other new joint prediction is connected to the root geom. Note that this is more challenging for objects like multi-drawer/door cabinets, because the LLM needs to infer which of the input OBBs is the root geom based on its sizes and relative orientation, but matters less for two-part objects like laptops.

---

### Official Review · Reviewer_S2Gc · 2024-11-10

**Soundness:** 3
**Presentation:** 3
**Contribution:** 3
**Rating:** 6
**Confidence:** 4

**Summary:**

The paper reconstructs articulated objects from visual observations. The approach utilizes a modular pipeline which first reconstructs part level geometry from segmentation and then uses a codegen LLM to combine the individual parts into an articulate assembled model to be executed in simulation. The paper compares to relevant recent baselines and demonstrate strong improvement. The approach also scales well to increasing number of joints due to its modular approach.

**Strengths:**

In my opinion, below are the strengths of the paper:

1. The paper scales well to increased number of joints. This has been a major limitation of preceding works and this work address it nicely with a modular approach i.e, part level reconstruction and code-gen integration for the subsequent steps.

2. Strong quantitative improvement numbers compared to recent state-of-the-art baselines, especially for increasing number of parts.

3. The presentation of the paper in nice and paper writing is easy to follow.

**Weaknesses:**

I have some question to the authors. In my opinion, below are the paper's weaknesses:

1. Why does the PARIS baseline struggle a lot? even for 2-part case? Did the authors try to tune their method? Based on the PARIS results' from the paper, it looks like it should reasonably work well for a simpler 2 part setting?

2. Despite good qualitative results, why are the resutls only shown on simpler objects like cupboards and laptop? Does the method work for varied articulated objects like scissors, stapler etc? Is this an inherent limitation of their method they only work for a subset of articulated objects for which they ahve a prior? If yes, that should be clearly stated as other baselines seems to work for more complicated articulated objects as well?

3. What is the timing result of the method? Some of the baselines mentioned i.e. CARTO, follow-up from CenterSnap [1] are very fast and don't require manual SAM prompting i.e. single-shot. This is not discussed very well in the related works.

4. I didn't find rigorous details on pretraining datasets for shape completion as well as datasets used for finetuning code llama. Those should be helpful to include. Also do authors plan to open-source their code? It looks like that will be helpful as well for the community to build up on?

[1] Irshad et al. CenterSnap: Single-Shot Multi-Object 3D Shape Reconstruction and Categorical 6D Pose and Size Estimation

**Questions:**

Please see the weakness section above for clarification questions. I look forward to seeing them in the rebuttal.

---

> ### Author Response · Authors · 2024-12-02
>
> Dear Reviewer S2Gc,
>
> Thank you for the detailed reviews and feedback. We hope the following additional clarifications, additional results, and the updated submission pdf will address your raised concerns.
>
> - Q1: Why is PARIS reported performance lower than the original paper.
> PARIS jointly optimizes object parts and the motion model based on a single rendering objective. As a result, their optimization is unstable and has large performance variances across different trials. While PARIS reports their best results across multiple trials in their paper, for fairness we report their average performance across 5 trials with random initializations.
>
> - Q2: how to handle more complex-shaped objects.
> A: Our pipeline can be divided into two parts: 1) geometry reconstruction (includes 1.1 part segmentation and 1.2 shape completion); 2) articulation prediction. We’d like to clarify that our method for 1) can handle arbitrary geometry shapes (e.g. scissors, faucet handles), if these objects are put into the training set for fine-tuning the part segmentation model and shape completion model. However, for 2), because we use the OBB formulation and select OBB edge as rotation center, our method can handle sliding joints (e.g. a sliding oven rack) but will be inaccurate for hinge joints where the joint is not overlapping with any OBB edge (e.g. scissors). We have updated the submission pdf to better clarify our assumption and the subsequent limitations. Since cuboid-shaped objects (cabinets, doors) are commonly seen, we still believe it’s of much value to reconstruct those objects, especially those with many parts, which our method handles much better.
>
> - Q3: Timing of our method.
> A: Our inference-time compute requirement is indeed larger than end-to-end methods like CARTO. To be specific: during SAM-prompting phase, we sample in 3D then projecting each 3D point onto 2D points across each single RGB image, hence the number of SAM forward pass scales linearly with the number of input camera views, however, we can cache the image embedding from SAM for every RGB image, and only call the light-weight prompt decoder for additional prompt points. Additionally, the inference compute for LLM code generation is dependent on the number of object parts, and roughly scales linearly with the generation token length. The shape-completion model forwarding is single-pass, which can be batched and takes <1min for all parts.
> Overall, our system is slower at inference time, but we deem it an affordable price to pay since this formulation can handle arbitrary numbers of object parts, and takes advantage of the strong generalization ability from pre-trained SAM models. CARTO-like model is single-shot on objects with one joint, hence would require either modifying the architecture for model output head, or running multi-round interaction to handle more object parts. We have updated the A.1 Discussion & Limitation section to better discuss these timing constraints of our method (see A.1.2), and also added the missing citation for CenterSnap.
>
>
> - Q4: Details on pre-training datasets & code availability.
> Please see Appendix 4.2 + 4.3 for more details on dataset preparation and model training details. We use a subset of PartNet-Mobility object assets and did our own RGB rendering and code conversion from raw URDF to OBB-relative MJCF files. Please let us know if you have further detailed questions. We will open-source code upon paper acceptance.

---

> > ### Comment · Reviewer_S2Gc · 2024-12-02
> > **Thanks for the response**
> >
> > Thanks for the response and clarifying my concerns. I don't have any additional questions.

---

### Author Response · Authors · 2024-12-04
**Summary of Response**

Thank you to all reviewers for taking the time to review. The reviewers raised several constructive feedback and questions that lead to insightful discussions. We therefore had hoped to respond with comprehensive answers supported with sufficient experiment results. This took up some time and led to reviewer mGLU lowering the score from 8 to 6. We would appreciate your understanding that the data preparation and model training/evaluating process is time consuming, and sincerely hope the score can be revisited after viewing our updated responses. We have provided clarifications, revisions in the updated submission pdf, and additional experiments. We summarize the main items below.

**1. Clarified assumption on object property.**

Our pipeline does both 1) geometry reconstruction and 2) articulation prediction. Our method for 1) can handle arbitrary geometry shapes. But for 2), our method can handle sliding joints (e.g. a sliding oven rack) but will be inaccurate for hinge joints where the joint is not overlapping with any OBB edge (e.g. scissors). To handle these objects, one possible extension is to add a regression MLP that takes in the OBB information and the LLM’s output for rotation direction as input, then predicts a more precise joint axis.

We have updated the submission pdf to better clarify our assumption and the subsequent limitations. Since cuboid-shaped objects (cabinets, doors) are commonly seen, we still believe it’s of much value to reconstruct those objects, especially those with many parts, which our method handles much better than prior work.


**2. Provided additional experiments on objects with more complex shapes.**

We have prepared a new dataset using two new object categories from PartNet-Mobility - Eyeglasses and Scissors, both have complex shapes beyond simpler, cuboid-like shapes (e.g. cabinets) that we evaluated on in the main paper. We use the new data to re-train both our proposed SAM fine-tuning model and shape-completion model, and test on held-out objects. These objects post additional challenges such as thinner shapes (e.g. blades), but we demonstrate that Real2Code can handle them with good output mesh quality. We provide visualizations for all the held-out test objects in this link:  https://sites.google.com/view/real2code-submission/complex-shapes

**3. Additional experiments on novel object categories**

To provide more insight into the zero-shot generalization ability of Real2Code, we have prepared 8 more test objects from Microwave and Door categories of PartNet (never seen in our training set). Many Microwave or Door objects have similar structural complexity to the StorageFurniture objects we trained on, but contains novel visual appearances and geometries. Therefore, our LLM module can generalize to these objects, but our fine-tuned SAM model and our shape completion model does not handle the OOD instances well.
We provide result visualizations for all the objects in this link:  https://sites.google.com/view/real2code-submission/generalization-to-novel-category

**4. Error propagation and main bottleneck of our system**

Overall, the main bottleneck of our system is the part-level segmentation quality, because as long as a valid part gets segmented out into a reasonable point-cloud, 1) the shape completion model is trained with noisy input such that it can complete the input; 2) it will result in a reasonable OBB as input to the LLM, which is essentially copying over one of the input edges as a joint axis. This is further evidenced by our additional experiment results above.

In the updated submission pdf, we have added a Discussion & Limitation section and ensure this limitation is well disclosed.
To further improve generalization, an interesting future direction is using SAMv2 to obtain more robust segmentations, or using large-scale pre-trained 3D generative models to complete the object shape. We provide a demo illustration of incorporating SAMv2 at this link: https://sites.google.com/view/real2code-submission/dataset-details

**5. Comparison to a more recent baseline**

We were asked to compare against DigitalTwinArt [1], a concurrent work that became available after our submission. We selected five multi-part objects with 2 or more moving parts for evaluation, and evaluate each object on 3 different seeds (i.e. a separate optimization run per seed), and report the average performance across 3 runs. We provide quantitative results and visualization of test objects in this link: https://sites.google.com/view/real2code-submission/comparison-with-digitaltwinart
Albeit not a full comparison, the results provide clear evidence that although DigitalTwinArt reported better performance than PARIS, it still struggles with objects with more than 1 moving parts and overall under-performs Real2Code.

[1] Yijia Weng, Bowen Wen, etc. Neural implicit representation for building digital twins of unknown articulated objects.

---

### Meta-Review · Area_Chair_upEw · 2024-12-13

**Metareview:**

This paper presents Real2Code, a novel approach for reconstructing articulated objects from visual observations by generating code, using fine-tuned LLMs specialized for this task. The writing is clear, the method is innovative, and the results are convincing and demonstrate the effectiveness of the proposed work. One limitation of the method is the dependence on SAM's segmentation capabilities. Particularly, Reviewer mGLU raises the concern about “the fine-tuned SAM's weak generalization performance, which struggles even with synthetic objects. The assumption of perfect segmentation in practical applications is unrealistic and substantially limits the system's scalability.” Nevertheless, this is a good paper that should be presented at the conference.

**Additional Comments On Reviewer Discussion:**

The rebuttal has addressed most of the reviewer concerns. Reviewer mGLU has a remaining concern that was not addressed by the reviewers (see meta-review), which is not critical for the acceptance of the paper, but which should be discussed by the authors in the limitations section.

---

### Decision · Program_Chairs · 2025-01-22

Accept (Poster)